# The Current Status of Photodynamic Therapy in Cancer Treatment

**DOI:** 10.3390/cancers15030585

**Published:** 2023-01-18

**Authors:** Wenqi Jiang, Mingkang Liang, Qifang Lei, Guangzhi Li, Song Wu

**Affiliations:** 1The Affiliated Luohu Hospital of Shenzhen University, School of Basic Medical Science, Health Science Center, Shenzhen University, Shenzhen 518000, China; 2Luohu Clinical Institute of Shantou University Medical College, Shantou University Medical College, Shantou University, Shantou 515000, China; 3Department of Urology, South China Hospital, Health Science Center, Shenzhen University, Shenzhen 518116, China

**Keywords:** PDT, photosensitizer, PDT-based combination therapy

## Abstract

**Simple Summary:**

Photodynamic therapy (PDT), a non-invasive cancer treatment strategy, has presented a broad scope in future clinical applications. This review mainly summarizes the advances of novel photosensitizers (PSs) characterized with high conversion efficiency and additional refreshing features. Besides, we also demonstrate the development of PDT-based combination therapies with chemotherapy, radiotherapy, immune therapy, and gene therapy and their outstanding neoplastic ablation efficiency.

**Abstract:**

Although we have made great strides in treating deadly diseases over the years, cancer therapy still remains a daunting challenge. Among numerous anticancer methods, photodynamic therapy (PDT), a non-invasive therapeutic approach, has attracted much attention. PDT exhibits outstanding performance in cancer therapy, but some unavoidable disadvantages, including limited light penetration depth, poor tumor selectivity, as well as oxygen dependence, largely limit its therapeutic efficiency for solid tumors treatment. Thus, numerous strategies have gone into overcoming these obstacles, such as exploring new photosensitizers with higher photodynamic conversion efficiency, alleviating tumor hypoxia to fuel the generation of reactive oxygen species (ROS), designing tumor-targeted PS, and applying PDT-based combination strategies. In this review, we briefly summarized the PDT related tumor therapeutic approaches, which are mainly characterized by advanced PSs, these PSs have excellent conversion efficiency and additional refreshing features. We also briefly summarize PDT-based combination therapies with excellent therapeutic effects.

## 1. Introduction

Since the late 1970s, photodynamic therapy (PDT) has gradually become an attractive technology for tumor treatment. It is approved for malignancy therapy by many countries such as the United States, Britain, France, Germany, and Japan [1]. Hermann von Tappeiner coined the term “photodynamic” in 1904 to distinguish photooxidation from the sensitization process in photography [2]. In 1903, the cancer cell killing derived from light irradiation with tumors coated with eosin was first reported. And then, numerous hematoporphyrin derivatives were subsequently applied for photodynamic therapy (PDT) [3,4]. In 1998, the US FDA approved photofrin for the treatment of early bronchial cancer and obstructive bronchial lung cancer [5,6]. Although this early laser-based anticancer photodynamic therapy demonstrates significant advantages, the first-generation photosensitizers, including hematoporphyrin derivaˉtive (HPD), dihaematoporphyrin ether (DHE), Porˉfimer sodium (Photofrin II), are inevitably encumbered by several drawbacks such as low photoconversion efficiency, non-targeted distribution, short wavelength light excitation (≤630 nm), and long metabolic clearance half-life (almost 4–6 weeks), all of which heavily hamper its clinical application. The first-generation PS (porfimer sodium), was the only PS approved and available for ablative procedure for superficial esophageal squamous cell carcinoma (ESCC) in Japan. However, due to the cost and side effects of PDT and development of endoscopic submucosal dissection (ESD), PDT lost its valuable role in clinical practice worldwide [7]. Hence, a large number of researchers are wild about working on boosting PS performances, they believe that improving the target specificity and enhancing the effective light penetration are crucial factors for the development of laser-based anticancer photodynamic therapy [8,9]. Thus, the second generation alternatives, including 5-ALA, meso-tetrahydroxyphenyl chlorin (m-THˉPC), methylene blue were found, and demonstrate approvable metabolic clearance efficiency, longer excitation wavelength (>630 nm), and tumor-targeted effect, greatly enhancing the clinical application prospect of PDT. The second-generation PS talaporfin sodium and diode laser was used as salvage therapy after chemoradiotherapy (CRT) or radiotherapy, showing good complete response (CR) rate without any phototoxicity, only with short sun shade period [7,10]. Furthermore, to meet better therapeutic performances, researchers are trying to combine PDT with other therapeutic strategies, including chemotherapy, radiotherapy, photothermal therapy, gene therapy, and immunotherapy [11,12,13,14,15].

During PDT, cytotoxic reactive oxygen species (ROS) are produced by PS excitation under specific wavelength light with oxygen (O_2_) supply to kill tumor cells. Among the critical parameters, the PSs and O_2_ concentrations are considered to be two critical factors for improving PDT performance. Furthermore, Different localization of photosensitizers in cells results in different PS efficiency. Studies have shown that high doses of PSs and high intensity laser can seriously cause cell damage and cell necrosis. For example, PSs related to plasma membrane can lead to oxidation reactions of proteins and phospholipids, which leads to rapid loss of membrane integrity and lysis [16,17]. In addition, low doses of PSs localized to the endoplasmic reticulum, lysosome and mitochondria can lead to apoptosis by directly damaging apoptotic proteins Bcl2 and Bcl-Xl [18]. It is now possible to optimize the performance of PSs by introducing heteroatoms, electron/energy effect-based, and aggregation-induced emission (AIE)-based strategies. Besides, numerous studies demonstrate that both tumor microenvironment (TME)-based exogenous oxygen supplementation and endogenous oxygen consumption regulations can significantly alleviate tumor hypoxia and remarkably enhance PDT efficiency. Tumor blood vessels are affected by oxidative stress under the action of PDT, leading to vasoconstriction, endothelial cell damage, thrombi formation, and eventually to intratumor blood flow stagnation, TME severe hypoxia and tumor necrosis [19]. In addition, methods for activating PSs have evolved from commonly photoactivation to different energy stimulations. For example, sonodynamic therapy, performing under ultrasonic activation with prominent penetration capacity, exhibits potent ablative effect on solid tumors. What’s more, studies have also found that PDT also plays a significant influence on intracellular proteins, DNA, and subcellular organelles, which facilitates the combination of PDT with other therapeutic strategies to inhibit tumor development synergistically. In this review, we briefly presented the PDT mechanisms and summarized the recent advances in PSs, O_2_ supply strategies, and PDT-based combined therapies in cancer treatment.

### 1.1. Mechanism of PDT

PS, O_2_, and excitation light sources are essential factors for PDT. There are three established scenarios for the energy level transition from S_1_ excited state to S_0_ singlet ground state: intersystem crossing to the lowest triplet state; giving off heat and fluorescence and thus going back to a rovibrationally excited S_0_ state; the triplet excited state can interact with some endogenous substances to form free radicals (such as hydrogen peroxide and superoxide anion) [20,21]. The excited states activate ROS formation in the following two ways: (Type I) the PS participates in electron transfer processes produce radicals and radical ions. PSs react directly with organic molecules by transferring hydrogen or electrons to form radicals. Besides, ROS such as superoxide anion (O_2_^−^), hydroxyl radical (OH.), and hydrogen peroxide (H_2_O_2_) can be created when free radicals combine with oxygen. (Type II) PS transfers energy from the triplet state to ground state molecular oxygen (^3^O_2_), creating highly reactive singlet state oxygen (^1^O_2_) [22,23,24,25] (Figure 1). Non-proportional reactions, such as the Haber–Weiss reaction, and Fenton reaction, can be fully utilized by the type I PDT process to offset the O_2_ consumption. Thus, the therapeutic efficacy for hypoxic malignancies is considerably enhanced [26,27,28]. By exposing the tumor tissue to long-wavelength light, Type II PDT is performed by excitation from the low-energy ground state (S_0_) to the high-energy excited state (S_1_) (650–900 nm, a wavelength range with excellent tissue penetration). ROS produced in PDT is involved in the oxidative damage to proteins, DNA, subcellular organelles, and blood vessels [29,30,31,32], which can also induce immunogenic cell death (ICD) and systemic immune response. In addition, PDT can also promote the secretion of a series of cytokines to improve antigen presentation, recruitment and infiltration of neutrophils and macrophages [33].

Different PSs are associated with various photodynamic therapeutic effects by diversiform mechanisms. AIE-active PSs could destroy lysosomes or mitochondria to induce apoptosis, necrosis, autophagy, or paralysis of target cells. Yin Li et al. found that cellular iron death could be caused by the oxidation of polyunsaturated fatty acids (PUFA) in lipid droplets (LD), which disrupt the balance between oxidative stress and antioxidants through excessive GSH consumption. Besides, the reduced glutathione peroxidase 4 (GPx4) expressions could also increase the accumulation of lipid peroxide (LPO), eventually leading to cellular iron death [34]. To obtain commendable and reliable PDT therapeutic effects, researchers are still working on developing and optimizing various PSs.

### 1.2. New Photosensitizers for PDT

Upon the activation of PS, abundant ROS is produced to destroy organelles, lipids, proteins, and nucleic acids, eventually inducing cell apoptosis, necrosis, or autophagy [35]. The characteristics, localization and excitation circumstances of PS, and the type of cells, have a significant effect on the therapeutic efficiency of PDT. The lowest triplet state of PS plays a vital role in PS excitation, but it remains a challenge to design efficient triple-excited PS with predetermined inter-systemic crossover properties, especially in the absence of heavy atoms [36]. Therefore, various approaches have been attempted to optimize PSs, such as the introduction of heteroatoms, strategies based on electron/energy effects, and AIE-based PSs. Among them, Rongcheng Han et al. introduced gold atoms into nanomaterials and prepared Gold Nanocluster (AuNCs) coated with dihydrolipoic acid (AuNC@DHLA). The AuNC@DHLA possesses 2~10^6^ GM, exhibits outstanding photodynamic characteristics and excellent two-photon (TP) optical properties. Additionally, compared to most conventional PS using the monoclinic oxygen (type II) mechanism, the type I process is used in the photochemical mechanism of AuNC@DHLA, which makes AuNC@DHLA become a more advantageous PS and improves its efficiency in vivo two-photon photodynamic therapy (TP-PDT) (Figure 2a) [37].

PSs activation can be achieved not only by light source but also by ultrasound. Song Wu et al. combined meso-tetra(4-carboxyphenyl) porphyrin (TCPP)-conjugated peroxidase (CAT) with fluorinated CS (FCS) to form TCPP-CAT/FCS nanoparticles (NPs) (Figure 2b), which synchronously achieves noninvasive excitation of PS in orthotropic bladder and tumor hypoxia amelioration triggered by the O_2_ production of tumoral endogenous H_2_O_2_ catalase, significantly enhancing the efficiency of sonodynamic therapy [38].

PSs characterized by aggregation-induced emission (AIE) exhibit remarkably enhanced fluorescence and photosensitivity. Wang Y et al. synthesized a PS (TPATrzPy-3+) with AIE performance and employed it to prepare an image-guided PDT agent. The PS is synthesized from two photochemically inert precursors, TPA-alkyne-2+ and MePy-N 3, in a Cu I-catalyzed click reaction generated by GSH-reduced MOF-199. After loading MOF-199 with TPA-alkyne-2+ and MePy-N 3, the precursor-loaded MOF-199 (PMOF) is coated with F-127 to obtain F-127-coated PMOF (PMOF NPs) (Figure 2c) [39]. Both in vitro and in vivo studies show that PMOF NPs could be effectively enriched into tumors after intravenous administration. Two precursors, TPA-alkyne-2+ and MePy-N 3, are then released from GSH-folded MOF-199 and effectively destroy tumor cells. Wu C et al. constructed AIE conjugated polymer with high Quantum yield (QY) in the NIR-II region using a dual fluorescence enhancement strategy. They employed phenothiazine and its derivatives (naphthothiadiazole or benzodithiadiazole) as the donor unit and synthesized a series of D-A conjugated polymers P1a~P1c, P2a~P2c, P3a~P3. Among them, because SP3 hybridizes sulfur and nitrogen atoms, the phenothiazine ring not only has excellent electron donating ability, but also presents a dihedral angle of 153° [40]. Tang et al. proposed a molecular design strategy to improve the brightness of conjugated polymers, namely, molecular structure planarization and torsional regulation. Four D-A conjugated polymers, PNIR-1, PNIR-2, PNIR-3, and PNIR-4, were synthesized, in which trianiline and tetraphenylethylene are used as electron donor units, and molecular rotors, while benzodithiadiazole is applied for electron acceptor units. Alkyl thiophene not only act as electron donor units and conjugated π-bridges, but also regulate the structure of polymer repeat units by providing the steric differences generated by different alkyl positions. Through the backbone structure regulation strategy to improve the brightness of polymer fluorophores, it provides a guiding idea for the development of high-brightness fluorescent dyes [41].

**Figure 2 cancers-15-00585-f002:**
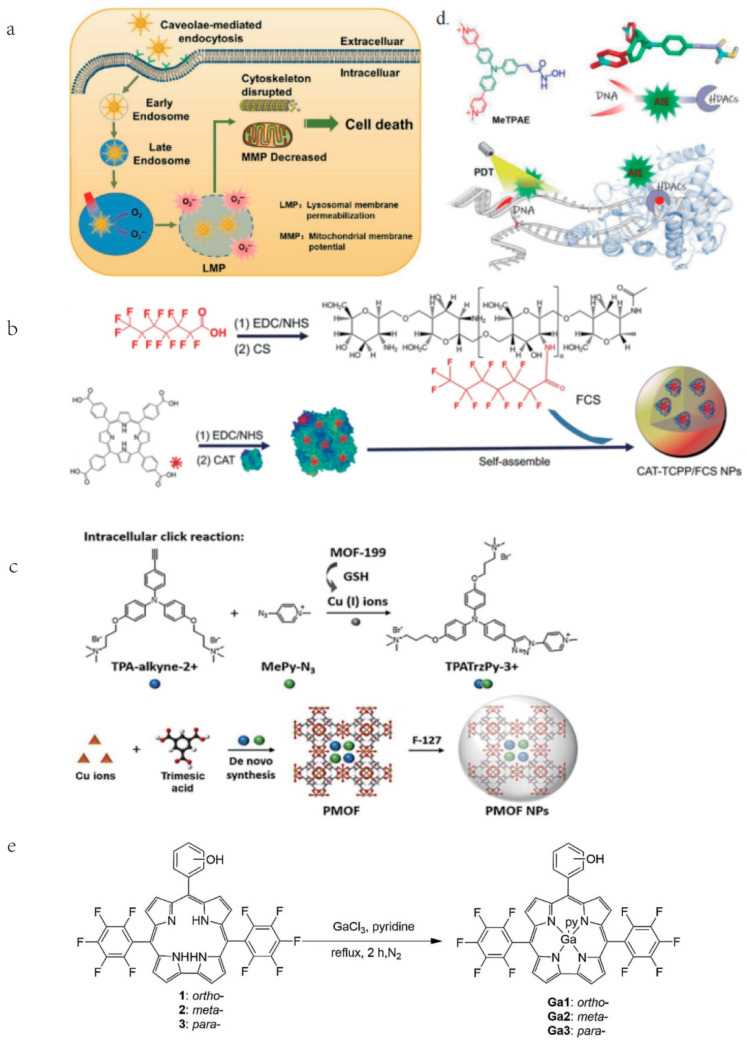
Different modifications and synthesis of different photosensitizers. (**a**) AuNC@DHLA can be internalized by endocytosis mediated by the lumen and accumulates in lysosomes, where LMP is produced as a result of ROS. Cell death eventually results from subsequent changes in MMP, mitochondrial morphology, and cytoskeleton disruption. Reproduced with permission from Ref. [37], Copyright (2020) American Chemical Society. (**b**) Synthesis of CAT-TCPP/FCS NPs. Reproduced with permission from Ref. [38], Copyright (2020) American Chemical Society. (**c**) Synthesis by using two photochemically inert precursors and the reduction of mof produced by GSH with copper(I) ions, TPATrzPy-3+ serves as an effective photosensitizer. Reproduced with permission from Ref. [39], Copyright (2021) John Wiley and Sons. (**d**) After MeTPAE enters the nucleus, it not only interacts with histone deacetylases (HDACs) to inhibit cell proliferation, but also precisely damages telomeres and nucleic acids through PDT. Reproduced with permission from Ref. [42], Copyright (2022) John Wiley and Sons. (**e**). The synthesis of freebase corroles 1–3 were synthesized though the one-pot procedures. The synthesis of Ga (III) corrole was accomplished by the reaction of freebase corrole with gallium (III) chloride in the presence of pyridine and N_2_ at 113 °C. Reproduced with permission from Ref. [43], Copyright (2020) Elsevier.

Kang-Nan Wang et al. prepared MeTPAE based on the triphenylamine structural backbone. MeTPAE performs its PDT effect by interacting with histone deacetylases (HDACs) and precisely disrupting telomeres and nucleic acids (Figure 2d), inducing cell cycle arrest and showing excellent PDT anti-tumor activity [42]. Hai-yang Liu et al. synthesized mono-hydroxy corroles 1–3, and their gallium (III) complexes Ga1–3 (Figure 2e), the single hydroxyl group on corroles can increase its amphipathic properties and facilitate cellular uptake capacity, thereby increasing its PDT activity. Meanwhile, Ga (III) corrole is the most commonly used metallocorrole for PDT that exhibits a more intense fluorescence and higher quantum yield than its freebase corrole. Corrole 3 and Ga3 are easily ingested by cells and localized in mitochondria and lysosomes, destroying mitochondrial membrane potential by increasing intracellular ROS levels, and inducing breast cancer cell death [43].

The above advanced PS provides a variety of strategies to overcome the predicament of the current PDT paradigm, which opens up promising paths for PDT in cancer therapy. Besides, there are also numerous attempts for combining PDT strategies with traditional therapeutic approaches that also exhibit excellent anticancer performances.

### 1.3. PDT Combined with Chemotherapy

Poor treatment outcomes of chemotherapy are caused by drug resistance, off-target effects, low bioavailability, being powerless for tumor heterogeneity, and severe systemic side effects [44,45] (Figure 3). Besides, cancer cells may take advantage of aneuploidy-induced genomic instability to change gene copy situations under selective stress conditions [46], resulting in malignant tumor progression [47,48]. It is well known that combination therapies using two or more treatment strategies could lessen toxicity and boost the therapeutic efficiency of a single therapeutic approach. Numerous non-invasive phototherapy (PDT and PTT) and chemotherapy are used in combination via drug delivery systems (DDSs) to obtain “increasing effect and decreasing toxicity” effects [49,50,51,52,53]. Precisely designed nanosystems show great therapeutic potential in the preclinical phase, it can not only increase drug bioavailability and biosafety but also allow a controlled drug release in TME [19]. Xuemei Yao et al. prepared a tumor extracellular and intercellular pH-stimulating DDS based on the dual pH response characteristic of mesoporous silica nanoparticles (MSN). They grafted histidine to the surface of MSN to form acid-sensitive polyethylene glycol tetraphenyl porphyrin zinc (Zn-PO-ca-PEG). The conjugated acid-sensitive cis-anhydride (CA) between Zn-POR and PEG separates at extracellular pH (~6.8), leaving the Zn-POR with a positive amino charge on its surface, remarkably promoting cellular internalization. In addition, in the intracellular acidic microenvironment (~5.3), the metal-supramolecular coordination disintegrates after Gatekeeper elimination, releasing the carried drug and Zn-Por [46]. Yang Zhao et al. reported that adoptive macrophage transfer led to a dramatically enhanced photodynamic therapy (PDT) effect of 2-(1-hexyloxyethyl)-2-devinyl pyropheophor-bide-alpha (HPPH)-coated polyethylene glycosylated nanographene oxide [GO(HPPH)-PEG] by increasing its tumor accumulation. Their self-degrading system (DOX) significantly improves pharmaceutics absorption, endosome escape capacity, and nuclear distribution and also shows the advantage of deeper tumor penetration and reduced off-target toxicity [54]. Zhou et al. prepared a photoactivated Pt (IV) prodrug polymeric PtAIECP. The chemotherapy drug doxorubicin (DOX) is wrapped in the PtAIECP to form the composited nanomaterial (PtAIECP@DOX), which have self-detection drug release performance and PDT-based synergistic therapy system. The effects of PtAIECP@DOX on prodrug activation, drug release, and synergistic therapy are verified in vitro and in vivo experiments [55].

Besides, chemotherapeutics combined with PDT could reduce the toxic side effects of chemotherapy alone. To cooperatively deliver chemotherapeutics and photosensitizers, Menglin Wang et al. loaded Ce6 by π-π stacking interaction with DOX and is connected to a personality polymeric carrier via ROS-sensitive terminal thione bonds (PEG-PBC-TKDOX). In this system, the ROS produced by PDT causes the nanocarrier to degrade and promotes drug release. Thus, non-targeted chemotherapeutics have minimal toxicity for their fixed release only where the excitation light is focused. This strategy is well suited to induce additional tumor growth inhibition with increased efficiency and lower toxicity [56]. It is also a suitable starting point for research into the use of PDT in conjunction with chemotherapy to lessen the side effects that seriously affect the prognosis of tumor patients.

### 1.4. PDT Combined with Radiation Therapy

Radiation therapy is a local treatment method for tumors. About 70% of cancer patients will undergo radiation therapy as part of their cancer treatment and commonly obtain approbatory therapeutic effects [57,58,59]. The effectiveness of radiotherapy depends on radiosensitivity, lesion location, and neoplasms types and so on. Among these, the proliferative cycle and pathological grading of tumors are closely associated with radiosensitivity. Additionally, the radiosensitivity of tumor tissues is also significantly influenced by O_2_ levels [60,61,62].

Similar to the combination of chemotherapy with PDT, remarkable synergistic therapeutic effects of PDT-based radiotherapy are widely confirmed. Liu Zhiyang et al. synthesized four polymers based on a triphenylamine-azafluorenone core that exhibit different photophysical properties and excellent biological applications. Cationization is an effective strategy to improve ROS generation and PDT efficiency of PSs. Cationized mitochondria-targeted PS shows a higher PDT efficiency than the non-ionized alternatives. Due to the AIE and ISC effects, the fluorescence intensity and ROS generation of AIE PSs increase simultaneously from the molecularly dispersed state to the aggregated state, which makes image-guided PDT possible without a complicated chemical process. In addition, several AIE PSs can also be employed as radiotherapy sensitizers. Compared to monotherapy, the combination of PDT plus radiation therapy dramatically boosts the efficacy of tumor ablation. This work provides a valuable exploration for improving PDT efficiency through the molecular design of PSs [63].

### 1.5. PDT Combined with Immunotherapy

PDT could efficiently trigger the release of tumor-associated antigens (TAAs) and danger-associated molecular patterns (DAMPs) from cancer cells, which would contribute to eliciting potent immune responses [33,64,65,66,67,68]. Yaxin Zhou et al. fabricated a PDT nanoplatform (CAM NP) loading AXT to enhance blood flow and alleviate the hypoxic microenvironment of tumor tissues, thus eventually enhancing the PDT therapeutic effects. The normalization of blood vessels also significantly facilitates immune cell infiltration, DC maturation, and T-cell activity. What’s more, the CAM NPs could increase the levels of tumor necrosis factor-α (TNF-a), interferon-γ (IFN-γ), and interleukins (IL-2 and IL-6), which further promote the tumor-infiltrating of T cells inducing prominently enhanced immune response. Their studies can boost levels of critical immune cytokines, promote DC maturation, activate immune effector cells, lead to remold tumor immunosuppressive microenvironment to produce a strong systemic immunological response [69].

Tang et al. reported a multifunctional nanorobot (NK@AIEdots) platform based on NK cell simulation, these robots can cross the blood-brain barrier to realize transcranial imaging and brain tumor treatment. In this study, the NK cell membrane is used to wrap AIE-conjugated polymer PBPTV to prepare NK@AIEdots, which can form a “green channel” to help NK@AIEdots cross the blood-brain barrier [70]. Considering the immune infiltration inhibition of lymphocytes (TILs) in the tumor microenvironment, Qinjun Chen et al. prepared a tumor-targeted PDT nanoplatform (Apt/PDGs@pMOF), these nanoparticles can promote cytotoxic T cell infiltration and simultaneously reconstitute the immunosuppressive microenvironment. In this research, Apt/PDGsˆs@pMOF demonstrates active tumor-target, PDT-triggered induction of PDG expression, ICD effect, DC maturation, and additional CTL penetration promoted by the Porphyrinic metal–organic framework (pMOF)-based PDT. Besides, the intratumoral MDSCs are selectively eliminated by penetrating PDG, thereby remolding the immunosuppressive tumor microenvironment (TME). Thus, in a bilateral 4T1 model, Apt/PDGsˆs@pMOF+L shows potent antitumor responses, dramatically inhibiting tumor growth [71]. Fang Sun et al. reported a prodrug, bromodomain-containing protein 4 inhibitor (BRD4i) JQ1, that counteracts C-myc and PD-L1 expression of tumor cell post PDT stimulation (Figure 4) [72]. Hongwei Cheng et al. synthesized glucose-PEO-b-PLLA-TEMPO to synergistically achieve antitumor immune activation. This nanoplatform simultaneously delivers the clinical therapeutic drug CUDC101 and the photosensitizer IR780, promoting M1 phenotypic polarization of TAM, IR780-mediated PDT, and CUDC101-triggered CD47 suppression [73]. Overall, PDT combined with immunotherapy may not only improve the activity of immune cells but also reduce the immune escape of tumor cells.

### 1.6. PDT Combined with Gene Therapy

Gene therapy has gained tremendous popularity in recent years, and many gene editing tools, such as Meganucleases (MNs), Zinc Finger Nucleases (ZFNs), CRISPR-Cas, and so on, have been applied to cancer therapy. It demonstrates better safety and tolerable adverse reactions than common chemotherapy and radiotherapy [74]. Although gene therapy has been well developed over the past three decades, most of its drugs are still in trials and rarely approved for clinical application. The biggest reason may be the ungovernable delivery efficiency and individual variation.

PDT has also been employed for collaborative treatment with gene therapy [75,76]. Lei Chen et al. designed a spherical nucleic acid (PSNA) for carrier-free and NIR light-controlled auto-delivery of siRNA and antisense oligonucleotides (ASO). A peptide nucleic acid-based ASO (pASO) and a NIR photosensitizer are found in the hydrophobic nucleus of PSNA, which also contains a hydrophilic siRNA shell (PS). On-demand disassembly of PSNA in tumor cells is made possible by the inclusion of a single heavy-state oxygen (^1^O_2_) cleavable linker between siRNA and pASO. The results demonstrate that the NIR PS produces ^1^O_2_ under near-infrared irradiation, which enables the disassembly and lysosomal escape of siRNA and ASO, and inhibits the hypoxia-inducible factor-1 (HIF-1) and B-cell lymphoma 2 (Bcl-2) expressions, eventually inhibiting tumor growth [77]. The DNA nanoscale sponges were successfully designed by Min Pan et al., which could effectively load and delivers PSs into tumor tissues, and significantly alleviate hypoxia-associated drug resistance [78].

The combination of PDT and gene therapy may have a better tumor-targeting effect, but the poor tissue penetration depth of light is still not completely solved. Aiming at this problem, Huaixin Zhao et al. reported the PDT nanoplatform containing persistent light-emitting nanoparticles (PLNPs) that activates glutathione at the tumor site without external laser excitation. PLNPs are employed as self-light sources and coated with MnO_2_ to prevent energy degradation. In response to the overexpressed glutathione in cancer cells, the MnO_2_ shell decomposes to provide Mn^2+^ and PDT essential O_2_. At the same time, PLNPs are released as a self-light source to activate PS and produce cytotoxic ^1^O_2_, which achieves tumor-specific PDT in McF-7 xenograft mice without external light exposure. In particular, the PDT properties of the nanocomposite can be initiated in GSH-overexpressed tumor cells and accomplished by self-oxygenation and self-irradiation, resulting in significant tumor inhibition [79].

Cu (I) 1, 2, 4-triazolate nanoscale coordination polymer (CP) multifunctional nanosheet is synthesized by a bottom-up technique. DNAzyme nanocarriers for gene therapy and endogenous photosensitizers for hypoxia-resistant TYPE I PDT can be carried by these CP nanosheets (PDT). Efficient loading and enzyme protection of DNA enzymes are fully achieved by [Cu (TZ)] nanosheets. As nanosheets react to GSH, the obtained DNAzyme/[Cu (TZ)] can preferentially activate DNAzyme in tumor cells, leading to targeted gene therapy of cancer cells [80]. Jinxing Huang et al. also used a gene-silencing approach to fuel cancer treatment. They proposed a nanotail therapy strategy to co-deliver the EGFR-TKI gefitinib (Gef) and YAP-siRNA for targeted drug/gene/PDT using large-scale dendritic polymeric nanoparticles. The resultant NPs could successfully escape from endosomes and lysosomes, ensuring the integration of Gef and YAP-siRNA into GEF-resistant non-small cell carcinoma (NSCLC) cells. The mechanism investigations demonstrate that nanoparticle can release Pyropheophorbide-a (Ppa) photosensitizer to produce PDT effect, suppress the activation of the YAP-AXL signaling pathway by releasing YAP-siRNA, and limit the EGFR transcription through Gef, significantly enhancing the PDT therapeutic efficiency [81]. Additionally, this nanomedicine combination can increase PDT sensitivity, and inhibit glycolysis by downregulating HIF-1α, which also lights up a new possibility for PDT-based gene therapy for tumor treatment.

Yu Liu combined mitochondria-targeted heptathiocyanine dye IR-68 with mitochondrial complex I and II inhibitor lonidamine (LND), and further assembled the mixture with albumin to form IR-LND@Alb NPs. This nano drug can target the mitochondria of tumor cells and activate the AMPK pathway by inhibiting mitochondrial oxidative phosphorylation (OXPHOS), thus down-regulating PDL1 expression to achieve the potent synergistic antitumor effect. Furthermore, IR-LND can down-regulate endogenous oxygen consumption to alleviate tumor hypoxia, thus providing enough O_2_ to enhance the therapeutic effect of PDT [82]. (Figure 5).

## 2. Discussion

The development and optimization of PS still require significant effort. To be further applied in the clinic, there are still many difficulties needed to be overcome. Currently, a variety of new PSs and PDT-based innovative therapies are emerging, which dramaticly improves the effectiveness and safety of PDT for tumor treatment. These innovative explorations mainly focus on the following three improvement strategies. Firstly, optimize PSs tolerant to hypoxia conditions and make it produces ROS efficiently in tumor anoxic environment. In addition, design tumor-targeted nanocarriers, and PSs activated by tumor microenvironment to improve the target capacity of PDT. Finally, increase the penetration depth of the excitation light for PS, such as chemiluminescence and bioluminescence strategies.

Different treatment approaches have their unique advantages and limitations. For example, the appropriate chemotherapy-phototherapy partition ratio in a combined chemotherapy-phototherapy system is critical for optimal cancer treatment outcomes. For the antibody-drug system, it is essential to enhance the targeting ability of therapeutic agents to improve the efficiency of combination therapy. For radiotherapy-phototherapy, the preparation of monodisperse nanomaterials with uniform size and high photon yield, combine nanomaterials with clinical radiation as sensitizers, and be effective at well-tolerated low radiation doses are still significant challenges. Some scholars even believe that PDT can be regarded as an intraoperative adjuvant therapy during the resection of some solid tumors (such as HCC). The fluorescence stimulated by PS will provide a visual aid to the surgeon. The optimal irradiation equipment and the most appropriate power and wavelength will be used to illuminate the cavity to kill the undetected residual tumors at the end of the surgery [83]. However, before considering the combination of PDT and surgery, we need to standardize PDT protocols in the clinic, including optimal PS and photo dose, optimal drug-optical interval, etc. It is also need to work on the development of irradiation devices specifically for open surgery and endoscopic resection.

With the synergistic action of immune drugs, PDT can remarkably enhance the ablation of local primary tumors and even efficiently inhibit tumor metastasis. The PDT dose, the intensity of the inflammatory response, and the release of the target cell antigen may all be related to the activation process or associated with the stimulation of immune cells. PDT can also be administrated with gene therapy to fuel the PDT therapeutic efficiency by precisely regulating target cell homeostasis at DNA, RNA, and (or) protein levels. Although PDT has shown great potential in cancer therapy, several issues that should not be ignored to achieve the goal of nanomaterial-based PDT for cancer treatment. Firstly, promoting biocompatible carrier materials to reduce side effects is the most critical factor for the PDT agent system. Besides, the unique Spatio-temporal form of monotherapy should be considered when combined with PDT collaborative therapy. Secondly, the main potential risks of the clinical application of nanoparticle-based PDT are systemic stability, the complexity of clearance, and long-term effects on the human body. In addition, the relationship between the immune system and PDT-based combination therapy remains fuzziness. More systematic safety studies must be conducted before these explorations applied in the clinic.

## 3. Conclusions

As a non-invasive therapy method, PDT has obtained significant progress in recent years. From early skin infection, and epidermal tumors, and now to solid tumor treatment, PDT especially shows excellent potential for cancer treatment. Photosensitizers play a crucial role in PDT, and numerous new multifunctional photosensitizers have emerged, that greatly promoting the application prospect of PDT. Besides, based on the noninvasive operation and unique therapeutic mechanism, PDT can be perfectly combined with chemotherapy, radiotherapy, immune therapy, and gene therapy. It is believed that with the continuous optimization of photosensitizers, and PDT-based combined treatment systems, PDT has a broader clinical application prospect in cancer treatment.

## Figures and Tables

**Figure 1 cancers-15-00585-f001:**
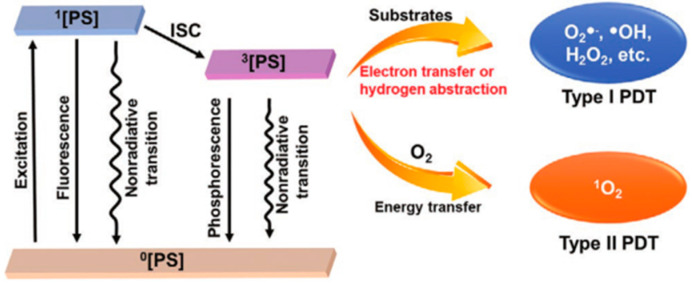
Basic PDT type I and type II process schematic diagram. Reproduced with permission from Ref. [27], Copyright (2021) John Wiley and Sons.

**Figure 3 cancers-15-00585-f003:**
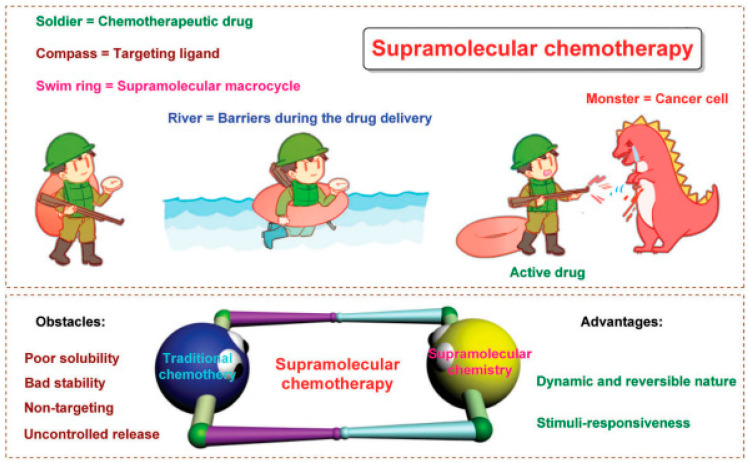
PDT and chemotherapy. Schematic diagram of supramolecular chemotherapy. Below is: Schematic diagram of supramolecular chemotherapy combining conventional chemotherapy and supramolecular chemistry. The obstacles to the clinical application of conventional chemotherapy and the advantages of supramolecular chemistry are pointed out. Reproduced with permission from Ref. [44], Copyright (2017) Royal Society of Chemistry.

**Figure 4 cancers-15-00585-f004:**
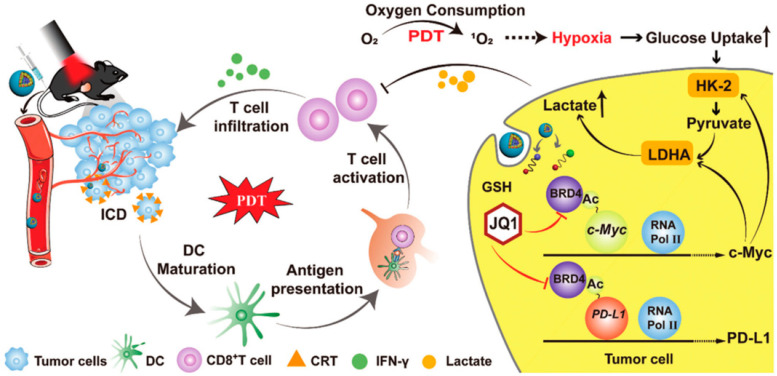
Mechanism of HCJSP-based combination immunotherapy for pancreatic tumors: PDT mediated by HCJSP-induced ICD in tumor cells, enhanced DC maturation, stimulated CTL activation, and resulted in tumor regression. Meanwhile, BRD4i JQ1 reduces the immunosuppressive tumor microenvironment and glycolysis stimulated by PDT by blocking the transcription of the c-Myc pathway downstream genes HK-2 and LDHA. Meanwhile, JQ1 specifically expressed down-regulated IFN-γ-induced PD-L1 surface tumor cell expression against PDT-inducible adaptive immune evasion. Reproduced with permission from Ref. [72], Copyright (2021) Sun Fang et al.

**Figure 5 cancers-15-00585-f005:**
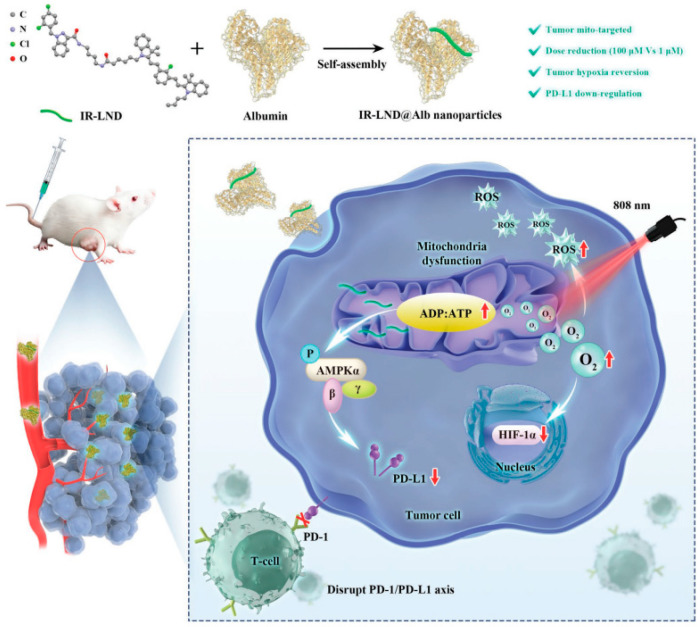
Combination therapy of gene therapy and PDT based on IR-LND@Alb: Nanoparticles reach mitochondria through the targeting of IR-68. LND inhibits mitochondrial oxidative phosphorylation (OXPHOS) and increases the ADP/ATP ratio. On the one hand, it can activate AMPKa channel and reduce the expression of PDL1 to prevent immune escape of tumors. On the other hand, it can alleviate the hypoxia in the tumor and generate ROS, thereby improving the photothermal efficacy of PDT and promoting the apoptosis of tumor cells. Reproduced with permission from Ref. [82], Copyright (2022) John Wiley and Sons.

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
