# Peer review of "The Current Status of Photodynamic Therapy in Cancer Treatment"

_cancers, 2023, doi:10.3390/cancers15030585_

Round 1

Reviewer 1 Report

The research was titled "Photodynamic Therapy in Cancer." This review examines the promising outcomes of various PDT-based materials developed for cancer treatment. Overall, I would recommend the publication in Cancers if the authors could please address my comment as well as the following minor points:

1.     Previously, a similar kind of title has been reported, therefore revised, the current trend as advised.

2.     In the introduction, it seems better to highlight briefly the various strategies that have been developed to improve the efficacy of PDT or, at the very least, those used to overcome the drawbacks of the PSs, knowing that there is currently a lot of advancement in this domain; in addition, include the new photosensitizers for PDT seems more interesting.

3.     What obstacles must be addressed in order to develop new photosensitizers, as stated in the introduction section?

4.     As suggested by the new subtitle, PDT is an essential source of photosensitizers, light, and oxygen, making it a critical PDT therapy.

5.     Discuss the new photosensitizer components and the results of using PDT in relation to the new photosensitizers.

6.     The conclusion part is where it is recommended that outstanding queries and potential directions for cancer PDT materials be considered.

7.     It is important to determine whether the type I or type II PDT process is more important for new photosensitizers in PDT cancer therapy.

Remarks

1.     Correct the mistakes such as spelling, superscript, and sentence: line no. 272,290,324,339,171, 94.

2.     All of the figures' numbers are missing from the explanation sentences. Insert the figure number, as suggested.

3.     The authors must discuss the recent studies concerning the new photosensitizer for PDT application in detail. The following paper can be used as references:(i) https://doi.org/10.1016/j.jconrel.2011.06.005, (ii) https://doi.org/10.1088/0957-4484/21/15/155103

Reviewer 2 Report

Title: Photodynamic therapy in cancer

In this study, the author provides a brief overview of PDT in cancer treatment. They summarized a large number of strategies for overcoming these existing limitations for PDT, application for multifunctional photosensitizers, as well as numerous PDT-based combination therapies. Based on above information, readers would have a systematic understanding of PDT, and can better carry out relevant researches.

The full text has rigorous logic, reasonable and well-founded, and is worthy of being accepted. But there are some minor flaws that can be improved:

1、          The article has partly elaborated on different materials at the present stage, but the innovation of nanomaterials and the shortcomings of the current research and the subsequent development are a bit shallow, and a summary and analysis of the current research problems can be added in the article.

2、          Some of the language in the article is a bit redundant; Can some redundant parts be removed to make the logic of the article clear?

3、          The article mentions the combination with gene therapy; Can the article briefly explain the current situation of gene therapy and the limitations of gene therapy?

4、          There are some abbreviations in the article that are not marked in English.

Reviewer 3 Report

This paper represents a possibly interesting review of the current status of photodynamic therapy (PDT) for treating cancer, together with a detailed discussion of different ways to improve PDT. Unfortunately the english language is often poor, making the paper's content hard or even impossible to understand. The paper should hence be completely rewritten by a native english speaker or equivalent, if possible with some understanding of the science involved. Also the authors should make a clear list of the abbreviations used at the beginning of the paper, and whenever an abbreviation is used for the first time in the text it should be explained. I will now list a number of remarks of concerns limiting myself to the first pages of the paper, so that the authors can see that a significant amount of work needs to be done in order to make their paper publishable in "Cancers":

1. In the abstract the authors mention higher "photo thermal" conversion efficiency when talking about PDT. This does not seem to make sense.

2. When discussing limited penetration, the authors should make it clear that they are talking about limited light penetration in human tissue.

3. In the introduction it would be reasonable when discussing the history of PDT to refer to an article by R.-M. Szeimies Akt. dermal. 2005, 31, 193-197.

4. In the introduction line 27, what is meant by "cancer benign disease therapy"?

5. In the introduction line 30, again it should read depth of light penetration. It is also not clear on line 30 what is meant by "non-selectivity for malignant tumour cells". presumably this is "lack of selectivity". 

6. The lack of selectivity is not uniquely caused by endogenous chromophores in healthy tissues. That may be the case to some extent in targeted therapies, but using typical porphyrin PSs there are many other reasons for selectivity and the lack of selectivity. The authors should be more specific in their descriptions.

7. An important argument that is not considered in the introduction is the lifetime of the PS in the human body. A number of excellent compounds have not made it through the FDA because of long-duration skin photosensitivity. mTHPC (Foscan) for instance.

8. On line 36 of the introduction you mention cytotoxic ROS. Now small quantities of ROS have a positive function in the body whereas large quantities of ROS have a cytotoxic effect. Thus ROS by themselves are not cytotoxic, only when present in large quantities.

9. There should be several references to AIE-based strategies in PDT of cancer. Non-luminescent molecules in solution can show bright emission upon aggregation. This may be due to the restriction of certain intramolecular motions. This needs at least a minimal explanation on how it is to be used clinically.

10. Line 59, "After exposure" should read "During exposure". After exposure there is no more light.

11. Line 60: "single linear state" should probably read "singlet state". It is not clear from your paper that the So singlet ground state, after absorbing light to the S1 singlet excited state, can now undergo 3 processes: 1. Intersystem crossing to the lowest triplet state (note, this is not a triplet excited state!), 2. Internal conversion via ro-vibrational relaxation back to So giving off heat, and 3. giving off fluorescence and thus going back to a rovibrationally excited So.

12. line 76: Please note that the important effect of PDT on blood vessels is the closure of these vessels. If the light exposure is taking place shortly after the iv injection of a PS, when the PS is still mainly located in the endothelial cells, the shape of the endothelial cells may change from elongated to rounded off, causing the tight junctions between the endothelial cells to break, and hence leakage of the blood and its components. This causes the release of van Willebrand factor and hence platelet aggregation and platelets sticking to the surface of the blood vessel, i.e. the clotting of the blood, thus depriving the tumor of oxygen and nutrients. The other (3rd) way in which PDT works on the tumors is by enhancing the iummune system. This important factor should also be discussed.

13. Line 80: It is not at all clear what is meant by paralysis of toxic ROS?

14. Line 94: one does not only induce apoptosis (though this may be desirable) but under slightly different conditions also necrosis and authphagy.

15. Line 96: "triple-excited state" should be the lowest triplet state. This state is generally attained by ISC from the vibrationally de-excited S1 state. Please note in Figure 1 that the arrow representing this should in principle be horizontal, and that this is followed by the ro-vibrational relaxation of the lowest triplet state (This is due to the Born-Oppenheimer approximation).

Round 2

Reviewer 3 Report

To properly correct this manuscript it should first be entirely rewritten by a native english speaker (or equivalent) with at least some knowledge of PDT. Besides the many faults in the english language which make the manuscript very hard to understand in many places, there are also multiple places in the manuscript where there are mistakes or just lack of precision. I have rewritten the simple summary, and I will give a few examples of other necessary corrections below (this for the time being is limited to the first few pages of the manuscript). Once the manuscript has been corrected extensively, and with care, I will be glad to go over it once again to correct the remaining errors in the detailed science.    
  1. The simple summary could for instance be changed as follows: "Photodynamic therapy (PDT), as a relatively novel, efficient, and minimally invasive cancer therapy, still presents some major challenges. The present paper reviews the new high-efficiency photosensitisers (PS) which have been developed to overcome the remaining shortcomings. We also review the possibilities of combining PDT with other therapies including chemotherapy, radiotherapy, immune therapy, etc.. PDT application scenarios are also summarised, as well as future PDT research possibilities."
  2. In the abstract: (a) please note that PDT is not new! (b) please replace "limited penetration" by "limited penetration of the light into the tissue to activate the PS" (c) please replace "non-selectivity" by "partial selectivity of the applied PS for the neoplastic tissue as compared to uptake of the PS in the surrounding normal tissue" (d) please replace " extreme oxygen dependence" by leaving out the word extreme (e) leave out "dramatically" (f) replace "discount" by "limit", (g) replace A by a, (h) replace "photo-thermal" by "photochemical", (i) replace "PDT systems" by "PDT dye molecules (PS)".
  3. In the introduction: For the historical part please refer to R.M. Szeimies in 2001, Comprehensive Series in Photosciences, Vol. 2, pp 3-15.
  4. Please also note that PDT was not first proposed by Hermann von Tappeiner et. al., as it already existed in prehistorical times; for instance it was used by the old Egyptians to cure vitiligo using a Psoralen containing extract and sunlight, several 1000 years ago.
  5. For hpd or hematoporphyrin derivative please refer to R.L. Lipson et. al. J. Natl. Cancer Inst. (1961) 26,1.
  6. Also in the introduction please replace "a long life in vivo" by "a long half life of the PS in the body, which in some cases may lead to prolonged skin photosensitivity".
  7. Also the expression "longer exposure to light" just by itself is meaningless.
  8. In the text you should also mention the third pathway after excitation from So to S1, which is internal conversion, i.e. electronic relaxation to the So ground state followed by ro-vibrational relaxation giving off heat.
  9. Line 46 overcame should be overcame
  10. Please also note that your expression "most photosensitisers (PS) are not toxic" is strange. PS toxicity depends on the dose applied.
  11. Please note that "cell death" after PDT is not only by apoptosis but can also be by necrosis or autophagy.
  12. etc,. etc. 

Round 3

Reviewer 3 Report

This paper needs to be entirely rewritten by a native english speaker or equivalent who has at least some knowledge of PDT.

Further examples of the very numerous corrections needed between lines 25 and 70.

Line 27: Limited light penetration

Line 42: von Tappeiner

Line 47: Replace Until by In

Line 49: anticancer photodynamic therapy

Line 56: target specificity

Line 57: effective light penetration

Line 58: anticancer photodynamic therapy

Line 61: monopolistic oxygen should be replaced by ROS (reactive oxygen species)

Line 66: Among the critical PDT parameters

Lines 66-67: the photosensitiser and O2 concentrations

Line 68: Furthermore different photosensitisers localise differently in cells leading to different PS efficiency ..

In this way I can go on correcting this paper, which as mentioned above is often not understandable or scientifically incorrect, mostly due to imprecise and faulty use of the english language. It is not my job to rewrite entirely this full review with so many errors.

I have already rewritten some of the early parts of this paper entirely, so please, have it completely rewritten by somebody who has a good command of the english language and also understands PDT, at least to some extent.

Author Response

Reviewer:

Comments and Suggestions for Authors:This paper needs to be entirely rewritten by a native english speaker or equivalent who has at least some knowledge of PDT.

Further examples of the very numerous corrections needed between lines 25 and 70.

1: Line 27: Limited light penetration.

2: Line 42: von Tappeiner

3: Line 47: Replace Until by In

4: Line 49: anticancer photodynamic therapy

5: Line 56: target specificity

6: Line 57: effective light penetration

7: Line 58: anticancer photodynamic therapy

8: Line 61: monopolistic oxygen should be replaced by ROS (reactive oxygen species)

9: Line 66: Among the critical PDT parameters

10: Lines 66-67: the photosensitiser and O2 concentrations

11: Line 68: Furthermore different photosensitisers localise differently in cells leading to different PS efficiency .

In this way I can go on correcting this paper, which as mentioned above is often not understandable or scientifically incorrect, mostly due to imprecise and faulty use of the English language. It is not my job to rewrite entirely this full review with so many errors.

I have already rewritten some of the early parts of this paper entirely, so please, have it completely rewritten by somebody who has a good command of the English language and also understands PDT, at least to some extent.

Reply: I’m sorry I made such a simple mistake.We have revised the details of the article according to your suggestions(line 21,35,39,,41,48,49,50-53,58,59,60-66.)This article has been rewritten by professionals who are proficient in English and have knowledge of PDT,Other changes are also marked by yellow notes.

Round 4

Reviewer 3 Report

There are too many errors left, both scientifically and in the english language to accept this paper in cancers. Sorry.
